# Quantum Tunneling and Complex Dynamics in the Suris’s Integrable Map

**DOI:** 10.3390/e26050414

**Published:** 2024-05-11

**Authors:** Yasutaka Hanada, Akira Shudo

**Affiliations:** 1Department of Information Science, Faculty of Arts and Sciences, Showa University, Yamanashi 403-0005, Japan; 2Department of Physics, Faculty of Science, Tokyo Metropolitan University, Tokyo 192-0397, Japan; shudo@tmu.ac.jp

**Keywords:** dynamical tunneling, integrable map, complex classical dynamics

## Abstract

Quantum tunneling in a two-dimensional integrable map is studied. The orbits of the map are all confined to the curves specified by the one-dimensional Hamiltonian. It is found that the behavior of tunneling splitting for the integrable map and the associated Hamiltonian system is qualitatively the same, with only a slight difference in magnitude. However, the tunneling tails of the wave functions, obtained by superposing the eigenfunctions that form the doublet, exhibit significant differences. To explore the origin of the difference, we observe the classical dynamics in the complex plane and find that the existence of branch points appearing in the potential function of the integrable map could play the role of yielding non-trivial behavior in the tunneling tail. The result highlights the subtlety of quantum tunneling, which cannot be captured in nature only by the dynamics in the real plane.

## 1. Introduction

Quantum tunneling is a subtle phenomenon because it cannot be captured by any power series, such as the WKB (Wentzel–Kramers–Brillouin) expansion, in terms of the Planck constant. This unique aspect can also be understood by recognizing that the power series expansion diverges, meaning that the tunneling effect is fundamentally non-perturbative [1].

Recent advances in resurgence theory have revealed that divergent series contain a wealth of information, even though they are divergent, and have provided a mathematical framework for dealing with exponentially small quantities [1,2,3,4,5]. Historically, the instanton method, proposed in the field theory and the theory of chemical reactions, is established when describing the tunneling effects of the system with a single degree of freedom [6,7,8,9,10,11,12,13,14,15,16,17,18,19,20]. The instanton is a path running along the imaginary time, so it is not the solution in the real plane. Along the same lines, it has long been recognized that it is necessary to consider *complex paths* in order to treat exponentially small effects. In resurgence theory, complex paths, including the instanton path, can be formulated as singularities in the Borel plane, which is obtained by applying the Borel transform to a given divergent series [1,2,3,4,5].

This paper discusses the subtlety of tunneling effects in a class of integrable maps for which one would expect nothing special since they are completely integrable. However, as shown below, this is not necessarily the case. The work presented here can be regarded as one of a series of studies on tunneling effects in nonintegrable systems based on the complex semiclassical analysis [21,22,23]. Since not only energetic barrier tunneling but also dynamical tunneling [24,25,26] are exponentially small effects, the importance of complex paths is obvious and the use of complex orbits is unavoidable if one intends to extract components originating from tunneling effects. The integrable map studied here shares a certain characteristic with nonintegrable maps, although it is integrable.

To study tunneling effects in nonintegrable systems, discrete-time dynamical systems, commonly referred to as maps, are often used as model systems. This is supported by several reasons, among which is that the behavior of complex paths has been well studied for discrete-time dynamical systems compared to continuous-time (Hamiltonian) dynamical systems [27,28,29]. However, we should keep in mind that it is not immediately clear whether the results derived for discrete-time dynamical systems, especially those based on complex dynamics, are applicable to continuous-time flow systems. The difficulty arises from the lack of a formal connection between the complex dynamics of the maps and continuous-time flow systems in which time is complexified.

On the other hand, one can find nonlinear integrable Hamiltonian flow systems, while nonlinear integrable maps are rather rare. The instanton is a path describing the tunneling transition in continuous Hamiltonian flow systems, and it is sometimes used as a reference when comparing the behavior of tunneling in integrable and nonintegrable systems [30,31,32]. In contrast, to the authors’ knowledge, there has not yet been an analysis of the tunneling effect for the map that is obtained by perturbing an integrable map. In most analyses, a continuous-time Hamiltonian system is taken as the integrable limit [33,34]. Such a treatment is somewhat self-inconsistent, and it is advisable to seek alternative solutions, if possible.

To fill this gap, here we study tunneling for a class of integrable maps found by Suris [35,36]. The integrability of the map is defined similarly to the case of continuous-time Hamiltonian systems. In particular, to verify the integrability of the two-dimensional symplectic map, it is sufficient to find a constant of motion under the discrete-time evolution. In Ref. [35], it was also shown that the maps with different types of potential functions provide ‘Hamiltonians’, and it is easy to check that they are conserved under the time evolution.

Here, we compare quantum tunneling for the integrable map and the associated ‘Hamiltonian’. We quantize the map as usual by introducing the one-step unitary operator, while Weyl quantization is applied to the continuous Hamiltonian. For a given classical Hamiltonian, recall that there are infinitely many quantizations resulting from the choice of the operator ordering. The differences are known to be of the order of $\hbar^{2}$, which is larger than exponentially small effects. Thus, even within quantizations of a continuous Hamiltonian, each of which is determined by the chosen operator ordering, tunneling effects are, in principle, uncontrollable. It is, therefore, not surprising that the difference in quantization between integrable maps and the associated integrable Hamiltonian systems can lead to different tunneling properties.

Our strategy in this paper is to perform high-precision numerical calculations of quantum systems and to observe the behavior of the complex dynamics generated by both the integrable map and the corresponding Hamiltonian flow. We compare the complex classical dynamics with the phase space profile of the time-evolved wave packet and reveal non-trivial signatures in the tunneling components, even though the map is integrable. We do not perform a semiclassical analysis here but explain significant features of the complex dynamics that lead to the semiclassical analysis in future studies.

Recently, the ultra-near integrable system has been studied in order to explore the difference in tunneling between integrable and nonintegrable systems [23,37]. Ultra-near integrable systems are defined as those systems in which the classical phase spaces do not exhibit any invariant structures inherent in nonintegrability in the size of the Planck cell. Even without islands of stability or chaotic seas, the tail of the wave function generates non-monotonic step structures [37], and the ergodicity of complex dynamics has been shown to play a key role in reproducing non-trivial tunneling behavior in nonintegrable maps [23]. Since there are no symptoms in the real phase space, the origin of the non-trivial tunneling tails comes down to the question of how chaos is born in the complex plane. The study of the integrable map is expected to be helpful in understanding this question.

The organization of this paper is as follows. In Section 2, we introduce the symplectic map which was found to be integrable by Suris [35], together with the Hamiltonian to which the orbits generated by the map are confined. We then define the unitary operator that provides the quantum dynamics for the map. Section 3 provides numerical results showing the difference in tunneling signatures between the integrable and the corresponding Hamiltonian system. Although the tunneling splitting exhibits only a tiny difference, the tunneling tails for the localized states constructed from the eigenfunctions forming the doublet differ significantly. In Section 4, we observe the behavior of the classical dynamics in the complex plane. This is only a preliminary study toward the semiclassical understanding of the non-trivial tunneling revealed in the integrable map, but the result provides important implications for understanding the observed phenomenon. What distinguishes the two systems is the fact that the solution of the Hamiltonian flow has no singularities in the complex plane, while the derivative of the potential function has branch points, which lead to multivalued dynamics in the integrable map. We also discuss issues that are expected to arise in the development of semiclassical analysis in the complex domain. Section 5 summarizes the results obtained in the present paper and provides some arguments, especially in relation to quantum tunneling in nonintegrable systems.

## 2. Model

To begin with, we introduce a two-dimensional area-preserving map $f:T^{2}\to T^{2}$,
(1)$$f:\left(\begin{array}{c} q_{k+1}\\ p_{k+1} \end{array}\right)=\left(\begin{array}{c} q_{k}+p_{k}-V^{\prime }\left(q_{k}\right)\\ p_{k}-V^{\prime }\left(q_{k}\right) \end{array}\right).$$
Suris [35] has shown that there exists a conserved quantity for the map:(2)$$H\left(q_{k+1},p_{k+1}\right)=H\left(q_{k},p_{k}\right),$$
for some specific choices of the potential function V(q). Among several possibilities, here, we take a potential function defined by [38]
(3)$$V_{\lambda }\left(q\right):=\frac{1}{\pi }\int_{0}^{q}\arctan\left(\frac{\lambda \sin2\pi q^{\prime }}{1+\lambda \cos2\pi q^{\prime }}\right)dq^{\prime }.$$
Note that the potential function $V_{\lambda }\left(q\right)$ is periodic with period 1. For this potential, the Hamiltonian
(4)$$H_{\lambda }(q,p):=-\cos2\pi p-\lambda \left[\cos2\pi q+\cos2\pi (q-p)\right],$$
satisfies the condition (Equation 2) for any λ. Below, we denote the map with the potential (Equation 3) by $f_{\lambda }$ to specify the map depends on the parameter λ. In this paper, the integration of $V_{\lambda }\left(q\right)$ is numerically evaluated by “double exponential quadrature” [39] implemented in Mathematica until numerical convergence is achieved to at least with 200 digits of mantissa.

Since the condition (Equation 2) holds for k∈Z, $H_{\lambda }$ can be regarded as a conserved quantity (Hamiltonian) for the map $f_{\lambda }$. In addition, the Hamiltonian flow $F_{\lambda }\left(q\left(t\right),p\left(t\right)\right)$ generated by the Hamiltonian equations of motion,
(5)$$\dot{q}\left(t\right)=\frac{H_{\lambda }(q,p)}{\partial p},\;\;\dot{p}\left(t\right)=-\frac{H_{\lambda }(q,p)}{\partial q},$$
is obviously integrable. Therefore, the map $f_{\lambda }$ does not exhibit chaotic motion and can be said to be integrable. In this paper, we refer to $f_{\lambda }$ as the Suris’s integrable map.

Figure 1a shows the functions $V_{\lambda }^{\prime }\left(q\right)$ and $V_{\lambda }\left(q\right)$ for different values of λ The profile of $V_{\lambda }^{\prime }\left(q\right)$ for |λ|<1 looks like a “skewed” sine function. As λ→±1, $V_{\lambda }^{\prime }\left(q\right)$ tends to a piecewise linear function, which is discontinuous at q=1/2 for λ=1 and q=0 for λ=−1. A fixed point (q,p)=(0,0) is elliptic-type if λ>0 and hyperbolic if λ<0. For |λ|>1, there are several discontinuous points in q∈[0,1). In this paper, we take λ∈(−1,0).

Figure 1b,c give phase space portraits for λ=−1/3 and −2/3. Each dot and solid curve of the same color represents a single orbit ${\left\{f_{\lambda }^{k}\left(q_{0},p_{0}\right)\right\}}_{k\in N}$ starting from an initial condition $\left(q_{0},p_{0}\right)$ and the corresponding contour curves of $H_{\lambda }(q,p)=H_{\lambda }(q_{0},p_{0})$, respectively. There are stable fixed points at (q,p)=(±1/2,0) and an unstable fixed point at (q,p)=(0,0). In addition, the map has stable periodic points with a period of 2 at $\left(q,p\right)=(\mp \arccos\left(-\frac{\lambda }{2}\right)/2\pi ,\pm \arccos\left(\frac{\lambda^{2}}{2}-1\right)/2\pi )$ and unstable periodic points with a period of 2 at (q,p)=(0,±1/2) (when we take mod1 for *q*). Separatrices starting from (q,p;E)=(0,0;−1−2λ) and (q,p;E)=(0,1/2;1) are shown in black dashed and black dash-dotted curves, respectively. As expected, the orbits ${\left\{f_{\lambda }^{k}\left(q_{0},p_{0}\right)\right\}}_{k\in N}$ for any $\left(q_{0},p_{0}\right)\in R^{2}$ lie on the contour curves of $H_{\lambda }\left(q,p\right)=H_{\lambda }\left(q_{0},p_{0}\right)$.

We next introduce the quantum dynamics associated with the classical map $f_{\lambda }$. Here, we assume the periodic boundary condition for both *q*- and *p*-directions, which leads to the finite-dimensional Hilbert space whose dimension is given by, say, *N*, and apply the canonical quantization to the one-step time evolution, which gives the quantum map $\hat{U}:|{\psi \rangle }_{k}\mapsto |{\psi \rangle }_{k+1}$ [40,41],
(6)$$\begin{array}{c} {\hat{U}}_{\lambda }:=e^{-\frac{i\;}{\hbar }{\hat{p}}^{2}}e^{-\frac{i\;}{\hbar }V_{\lambda }\left(\hat{q}\right)}. \end{array}$$
Floquet theorem leads to the eigenvalue equation,
(7)$$\begin{array}{c} {\hat{U}}_{\lambda }|\psi_{n}\rangle =e^{-\frac{i\;}{\hbar }E_{n}}|\psi_{n}\rangle . \end{array}$$
Correspondingly, we introduce the quantized Hamiltonian ${\hat{H}}_{\lambda }(\hat{q},\hat{p})$ by adopting the canonical quantization for the classical Hamiltonian (4). Note here that we take the Weyl’s ordering for the product of *q* and *p*. The eigenvalue equation for ${\hat{H}}_{\lambda }$ then reads
(8)$$\begin{array}{c} {\hat{H}}_{\lambda }|\Psi_{n}\rangle =E_{n}|\Psi_{n}\rangle . \end{array}$$ In the following, the quantum number *m* of the quasi-eigenstate $|\psi_{m}\rangle$ is determined by the quantum number of the eigenstates $|\Psi_{n}\rangle$ attaining the maximal overlap $|\langle \psi_{m}|\Psi_{n}\rangle {|}^{2}$.

In this paper, numerical computations for the quantum map are performed by using arbitrary precision arithmetics implemented in Mathematica, MATLAB 5.2.5.15470, with the ADVANPIX toolbox  [42].

## 3. Discrepancy between the Wave Function $|\Psi_{L}\rangle$ and $|\psi_{L}\rangle$

The discrete map $f_{\lambda }$ and the continuous Hamiltonian flow $F_{\lambda }$ share the same constant of motion $H_{\lambda }\left(q,p\right)$, but the quantization procedures are different: the former is quantized by introducing the single-step unitary operator ${\hat{U}}_{\lambda }$, while Weyl’s quantization is applied to the Hamiltonian $H_{\lambda }$. Thus, it is by no means obvious whether the eigenvalues and eigenfunctions are identical and have the same properties.

In particular, we are interested here in tunneling effects, so the difference in quantization can be crucial. This is because, as mentioned above, exponentially small effects slip through ordinary semiclassical arguments, which are made by expressing related quantities in terms of the expansion of *ℏ*. Even within quantizations for the Hamiltonian $H_{\lambda }$ there is ambiguity about the operator ordering, leading to an uncertainty of the order of *O*(*ℏ*^2^), so it is all the more unclear what will happen if the quantization method differs from each other.

Here, we examine tunneling splittings generated in the nearly degenerate energy levels of the quantum map (6). To obtain a double-well-like setting, we impose the periodic boundary condition on the phase space region (q,p)=[−1,1)×[−1/2,1/2). The periodicity of V(q) leads to the point-symmetric phase space around (0,0) (see Figure 1), and the ‘libration’ motion appears for $-1+2\lambda <H_{\lambda }(q,p)<-1-2\lambda$. Due to the symmetry with respect to the origin, the energy levels of the quantum map produce tunneling splittings, similar to the one-dimensional double-well potential system.

Figure 2a shows some eigenfunctions $\psi_{n}\left(q\right)$ and $\Psi_{n}\left(q\right)$ for ℏ=1/200π in semi-logarithmic scale, and Figure 2b shows the corresponding Husimi representation for $|\Psi_{n}\rangle$ in normal scale. There is no noticeable difference between $\psi_{n}\left(q\right)$ and $\Psi_{n}\left(q\right)$, and the Husimi representation has support on the contour curves $H_{\lambda }(q,p)=E_{n}$. Figure 3a(i,ii) plot the tunneling splitting defined by
(9)$$\Delta E_{0}=\left|E_{1}-E_{0}\right|,\;\Delta E_{0}=E_{1}-E_{0},$$
as a function of 1/ℏ and −λ, respectively. In contrast to the almost perfect agreement of the eigenfunctions shown in Figure 2a, we can find a few orders of magnitude difference between the tunneling splittings $\Delta E_{0}$ and $\Delta E_{0}$ [see also Figure 3b(i,ii)], while the parameter dependence of $\Delta E_{0}$ and $\Delta E_{0}$ on 1/ℏ and −λ is qualitatively the same; both are expected from the semiclassical analysis for integrable systems.

A tiny difference in the magnitude implies that some discrepancy should be hidden in the eigenfunctions. Figure 3c plots the magnitude $1-|\langle \psi_{n}|\Psi_{n}\rangle {|}^{2}$ as a function of −λ, which shows that the difference between $|\psi_{n}\rangle$ and $|\Psi_{n}\rangle$ does indeed exist, although it is exponentially small, less than $10^{-8}$ for λ∈(−1,0). Note also that the difference $1-|\langle \psi_{n}|\Psi_{n}\rangle {|}^{2}$ increases as −λ tends to 1.

The tunneling splitting appears as a result of the overlap of the wave function localized at the left (q<0) and the wave function localized at the right (q>0):(10)$$\begin{array}{c} |\psi_{L(R)}\rangle :=\frac{1}{\sqrt{2}}(|\psi_{1}\rangle \pm |\psi_{0}\rangle ),\;|\Psi_{L(R)}\rangle :=\frac{1}{\sqrt{2}}(|\Psi_{1}\rangle \pm |\Psi_{0}\rangle ). \end{array}$$
Figure 4a,b present the wave functions $\Psi_{L}\left(q\right)$ and $\psi_{L}\left(q\right)$ and the corresponding Husimi representation, respectively. Let $q_{-}^{\times }$ and $q_{+}^{\times }$ be the turning points of the manifold $H_{\lambda }(q,p)=E_{0}$ projected onto the *q*-axis. As seen in Figure 4a, $|\Psi_{L}{\left(q\right)|}^{2}$ has an exponentially decaying tunneling tail in the region $q\in (q_{-}^{\times },q_{+}^{\times })$, as predicted by the WKB argument.

The Husimi representation provides more detailed information. As shown in Figure 4b(i), the amplitude of the Husimi representation for $|\Psi_{L}\rangle$ decays exponentially along the instanton curve. The instanton curve is obtained here by the purely imaginary time evolution t∈iR of the Hamiltonian flow $F_{\lambda }$ starting from the turning point $q_{-}^{\times }$.

$|\psi_{L}{\left(q\right)|}^{2}$ also shows an exponentially decaying tunneling tail from $q=q_{-}^{\times }$ down to q=0. However, $|\psi_{L}{\left(q\right)|}^{2}$ stops decaying just beyond q=0 and then forms a plateau with an oscillatory pattern in the region q>0. As shown in Figure 4b(ii), the Husimi representation of $|\psi_{L}\rangle$ tells us that the localization along the separatrix starting at (q,p)=(0,0) is responsible for the oscillatory plateau observed in the plot of $|\psi_{L}{\left(q\right)|}^{2}$.

The plateau found in $|\psi_{L}{\left(q\right)|}^{2}$ and the localized component along the separatrix are reminiscent of “tunneling across the separatrix”. Note that its importance for our understanding of the anomalous enhancement of the tunneling probability has been pointed out in Ref. [34]. However, they are not necessarily the same phenomenon. The main difference is that similar oscillation patterns in nonintegrable systems penetrate into the region q<0, while the plateau found in the Suris’s integrable map does not. To confirm this, we examine the interval $\left[q_{-}^{★},q_{+}^{★}\right]$ in which $|\psi_{L}{\left(q\right)|}^{2}$ is 10 times larger than $|\Psi_{L}{\left(q\right)|}^{2}$. Here, the edges $q_{-}^{★}$ and $q_{+}^{★}$ are defined by
(11)$$\begin{array}{cc} q_{-}^{★} & :=\min\left\{q\in \left[-1,1\right):\log_{10}{\left|\frac{\psi_{L}\left(q\right)}{\Psi_{L}\left(q\right)}\right|}^{2}>1\right\}, \end{array}$$
(12)$$\begin{array}{cc} q_{+}^{★} & :=\max\left\{q\in \left[-1,1\right):\log_{10}{\left|\frac{\psi_{L}\left(q\right)}{\Psi_{L}\left(q\right)}\right|}^{2}>1\right\}. \end{array}$$

Figure 5 plots $q_{-}^{★}$ as a function 1/ℏ. Dots and a black dashed curve in Figure 5 indicate numerical data from $q_{-}^{★}$ and a linear fit to the obtained data, respectively. The plot shows that the value of $1/q_{-}^{★}$ increases linearly with 1/ℏ, meaning that $q_{-}^{★}\to 0$ for 1/ℏ→∞. Note in particular that the decay rate of $1/q_{-}^{★}$ with 1/ℏ is extremely slow compared to nonintegrable cases [34].

## 4. Toward Semiclassical Understanding for Plateau with Oscillatory Pattern

While the WKB (semiclassical) analysis for eigenstates of integrable Hamiltonian systems is well established [43], there is no semiclassical prescription to analyze eigenstates of nonintegrable systems. On the other hand, the semiclassical analysis in the time domain is available even for nonintegrable systems, and the methodology incorporating the complex domain is now in a usable state [22,44,45,46,47]. Here, we discuss the origin of the plateau accompanied by oscillations observed in the quantum Suris’s integrable map by taking the time-domain semiclassical approach.

Among possible options for initial conditions, we choose state $|\Psi_{L}\rangle$ since we are particularly interested in the discrepancy between $|\psi_{L}\rangle$ and $|\Psi_{L}\rangle$. Figure 6a,b illustrate the propagation of the wave function ${\hat{U}}_{\lambda }^{k}|\Psi_{L}\rangle$ in the *q*-representation and the Husimi representation, respectively. We can see that the amplitude of the wave function ${\hat{U}}_{\lambda }^{k}|\Psi_{L}\rangle$ for $q\in [q_{-}^{★},q_{+}^{★}]$ increases rapidly with time, and after four steps, it already reaches a profile similar to $\psi_{L}\left(q\right)$. Note that the saturated amplitude of ${\hat{U}}_{\lambda }^{k}|\Psi_{L}\rangle$ in the interval $q\in [q_{-}^{★},q_{+}^{★}]$ is slightly higher than the amplitude of $\psi_{L}\left(q\right)$.

The Husimi representation shows us an explicit wave packet propagation in phase space. As time evolves, the dominant part of the wave function ${\hat{U}}_{\lambda }^{k}|\Psi_{L}\rangle$ passes through the fixed point (q,p)=(0,0) and propagates along the separatrix. After five steps, the amplitude spreads almost uniformly along the separatrix. This tells us that the plateau with oscillation observed in $\psi_{L}\left(q\right)$ appears as the result of the tunneling transport along the separatrix associated with the fixed point (q,p)=(0,0).

### 4.1. Complex Dynamics of $F_{\lambda }$ and $f_{\lambda }$

In this section, we explore the classical dynamics $F_{\lambda }$ and $f_{\lambda }$ in the complex domain. To do this, we introduce notations for the real and imaginary parts as *q* = Re *q* + i(Im *q*), and *p* = Re *p* + i(Im *p*), respectively.

First, we observe the solution curves in the complex plane under the flow $F_{\lambda }$. They lie on the equi-energy surface specified by the condition $H_{\lambda }=E_{n}\in R$. To obtain these curves, we first find the turning points $q_{\pm }^{\times }$, marked by the black “×” in Figure 7, and then solve the equations of motion for the Hamiltonian $H_{\lambda }$ along the imaginary time axis iR to connect the turning points $q_{\pm }^{\times }\in R$ Recall that this solution curve, shown by the thick black curve, is nothing more than the instanton. Along the instanton curve, we place equally spaced initial conditions, from each of which we solve the equations of motion along the real-time axis R, yielding a set of closed curves shown in green.

As for the dynamics of the map $f_{\lambda }$ in the complex plane, the multivaluedness of the potential function $V_{\lambda }^{\prime }(q)$ generates non-trivial behavior. To see this, we focus on the dynamics at the points (Re *q*, Re *p*) = (0,0), (0, ±1/2), and (±1/2, ±1/2), respectively. The equations of motion of the Hamiltonian flow $F_{\lambda }$ are written explicitly as
(13)Imq˙(t)=2πα1sinh2πImp+α2λsinh2π(Imq−Imp),Imp˙(t)=−2πλα3sinh2πImq+α4sinh2π(Imq−Imp),Req˙(t)=Rep˙(t)=0,
where
$\left(\alpha_{1},\alpha_{2},\alpha_{3},\alpha_{4}\right)=\left(1,-1,1,1\right)$ for (Req,Rep)=(0,0),(α1,α2,α3,α4)=(−1,−1,−1,1) for (Req,Rep)=(±1/2,±1/2), and $\left(\alpha_{1},\alpha_{2},\alpha_{3},\alpha_{4}\right)=\left(-1,1,1,-1\right)$ for (Req,Rep)=(0,±1/2). Since Re *q*(*t*), Re *p*(t) = *const.*, the solution curves shown in Equation (13) are restricted to the (Im *q*, Im *p*)-plane. Note that the fixed points (Im *q*, Im *p*) = (0, 0) of Equation (13) are all hyperbolic, and the right-hand sides of Equation (13) consist of monotonic and entire functions so that the solution curves around (Im *q*, Im *p*) = (0, 0) diverge monotonically to Im *q*(*t*), Im *p*(*t*) → ±∞ under forward time evolution (see Figure 8).

The green dots in Figure 7 show the classical orbits ${\left\{f_{\lambda }^{k}\left(q_{0},p_{0}\right)\right\}}_{k\in Z}$ for which the initial points $\left(q_{0},p_{0}\right)$ are set along the instanton. We observe that all the orbits ${\left\{f_{\lambda }^{k}\left(q_{0},p_{0}\right)\right\}}_{k\in Z}$ preserve the energy *H_λ_* = *E* as expected, but there are exceptions: the orbits starting at (Req0,Rep0)=(0,0) are not bounded on the Hamiltonian flow $F_{\lambda }$. As explained below, the multivaluedness of the function $V_{\lambda }^{\prime }\left(q\right)$ for q∈C leads to such anomalous behavior. In Figure 7b, the cyan dots show the projection of the itinerary of the orbit whose initial condition is placed on the instanton orbit onto the (Req,Rep)-plane. We can see that the real part (Reqk,Repk) of the orbit jumps among (Reqk,Repk)=(0,0), (±1/2,±1/2), and (0,±1/2), while the real part (Req(t),Rep(t)) does not change under the Hamiltonian flow $F_{\lambda }$ [see Equation (13)].

This itinerary of the orbit can be understood by considering the derivative $V_{\lambda }^{\prime }\left(q\right)$ of the potential function in the complex *q*-plane. Introducing the new variable,
(14)$$\begin{array}{c} z=\frac{\lambda \sin2\pi q}{1+\lambda \cos2\pi q}\;, \end{array}$$
we find that the derivative $V_{\lambda }^{\prime }\left(z\right)$ can be expressed as
(15)$$\begin{array}{c} V_{\lambda }^{\prime }\left(z\right)=\frac{1}{\pi }\arctan z\;, \end{array}$$
where the arctangent function has an alternative expression,
(16)$$\begin{array}{c} \arctan z=\frac{1}{2i\;}\log\left(\frac{1+i\;z}{1-i\;z}\right)=\int \frac{dz}{1+z^{2}}\;, \end{array}$$
which obviously has the branch points in the *z*-plane at z=±i, originating from the logarithmic function. Correspondingly, in the *q*-plane, $V_{\lambda }^{\prime }\left(q\right)$ has the branch points at $q=m\pm i\;\frac{1}{2\pi }\log\left(-\lambda \right)=:m\pm i\;q^{⋄}$, where m∈Z (see Figure 9).

Now, we consider the itinerary of the orbit starting from (Req,Imq,Rep,Imp)=(0,Imq0,0,Imp0). In other words, we set the real part of the initial points to (Req0,Rep0)=(0,0), but take various values for the imaginary parts (Imq,Imp)=(Imq0,Imp0), assuming that |Imq0|<q⋄. Since the $V_{\lambda }^{\prime }\left(q\right)$ is a multivalued function in the logarithmic type, the one-step iteration leads to an infinite number of images. Several studies have explored the complex dynamics associated with multivalued functions [48], but here we consider only the one-step iteration to get rid of the complication caused by the multi-step iteration.

Since the stability at the origin (Imq,Imp)=(0,0) is hyperbolic (see Figure 8a), for any given Imp0, there exists an initial condition |Imq0|<q⋄ such that the next step satisfies the condition |Imq1|>q⋄. Recall that the Hamiltonian flow follows Equation (13), so in the time evolution, the Req(t) remains constant (Req(t)=0), and only the value of Imq(t) changes. Therefore, in the (Req,Imq)-plane, the point (0,Imq0) passes over the branch points $\left(0,\pm q^{⋄}\right)$ before and after being mapped to the point (0,Imq1). After passing one of the branch points, the mapped point gains either 1/2+m and $-1/2-m^{\prime }$$\left(m,m^{\prime }\in Z\right)$, reflecting the multivalued nature of the potential function (15). As a result, the initial point (Req0,Rep0)=(0,0) is mapped to the lattice points $\left(1/2+m,1/2+m^{\prime }\right)$$\left(m,m^{\prime }\in Z\right)$. However, these lattice points can be identified as (1/2,±1/2) and (−1/2,±1/2) by applying the periodic boundary condition. Therefore, the real parts (Req0,Rep0)=(0,0) are mapped to (Req1,Rep1)=(1/2,±1/2) and (−1/2,±1/2).

For the initial points (0,Imq0) in the (Req,Imq)-plane that do not pass the branch points within the single-step iteration, they simply move along the solution curve governed by Equation (13). The points generated by the multivalued nature of the map are again identified by applying the periodic boundary condition. Depending on the initial condition (Req0,Rep0), the single step iteration in the (Imq,Imp)-plane gives the map from (Imq0,Imp0) to (Imq1,Imp1) on the same flow curve. Note that the transition between different (Req,Rep)-planes does not occur because we assume here that the initial points (0,Imq0) in the (Req,Imq)-plane do not pass the branch points.

### 4.2. One-Step Propagation

As illustrated in Figure 6, the plateau with oscillation observed in the wave function $|\psi_{L}\rangle$ is reproduced by a wave function with relatively short time steps. This should make it possible to develop the semiclassical analysis in the time domain.

As explained in the previous section, the multivaluedness of the map leads to the transition that never occurs in the integrable Hamiltonian flow. The orbits of the Hamiltonian flow $F_{\lambda }$ starting from the unstable fixed point (Req(0),Rep(0))=(0,0) do not change the real part, i.e., (Req(t),Rep(t))=(0,0)(t>0) independent of the initial conditions (Imq0,Imp0), while the orbits generated by the map $f_{\lambda }$ jump from (Re *q*_0_, Re *p*_0_) = (0, 0) to either (Re *q*_1_, Re *p*_1_) = (1/2, ±1/2) or (−1/2, ±1/2) when they pass over the branch points, otherwise they stay at (Re *q*_1_, Re *p*_1_) = (0, 0). Such a transition should explain the propagation of the wave function along the separatrix, but some issues need to be clarified before proceeding.

The first one is that the orbit in the complex plane is bifurcated even in one-step propagation, reflecting the multivaluedness of the map. The transition from (Req,Rep)=(0,0) to (1/2,1/2) is observed in the one-step propagation of the quantum wave packet [see Figure 10], while the orbit can jump from (Req,Rep)=(0,0) to (±1/2,∓1/2) as explained in the previous section. We expect that this can be solved by properly treating the Stokes phenomenon when performing the saddle point approximation. If the path from (Req,Rep)=(0,0) to (±1/2,∓1/2) gives rise to a divergent solution, then it should be removed from the final solution due to the Stokes phenomenon. For multi-step analysis, however, the multivaluedness of the map must be treated in each step, which makes semiclassical analysis difficult. Therefore, we should first perform a single-step semiclassical calculation. As can be seen from Figure 6, the plateau along the separatrix becomes more visible as time evolves. It may be a relatively short time, but we have to perform a semiclassical calculation with multiple time steps. This requires a more sophisticated approach to the Stokes phenomenon.

On the other hand, Figure 10 shows that the plateau can also be made visible by increasing the parameter −λ. This may give us the impression that the one-step semiclassical analysis is sufficient to explain the observed phenomenon, and one can get rid of multi-step calculations. However, the problem is not so simple, because as the parameter −λ is increased, the potential $V_{\lambda }\left(q\right)$ and its derivative $V_{\lambda }^{\prime }\left(q\right)$ become sharper, as shown in Figure 1a. This reminds us of the diffraction effect [49], which is expected to occur when the potential function has discontinuities. We should include higher-order terms in the saddle approximations, or more legitimately develop uniform approximation-type arguments to cope with such situations.

## 5. Conclusions and Discussion

In this paper, we have studied quantum tunneling for a nonlinear integrable map found by Suris, with a particular focus on the comparison with the corresponding continuous Hamiltonian system. Although the Suris’s map is completely integrable in the sense of Liouville–Aronld in Hamiltonian systems, and the orbits in the real plane generated by the Suris’s map lie on the equi-energy curve for the continuous Hamiltonian, our numerical computations performed with arbitrary precision arithmetic have revealed that the characteristics of the tunneling tails in the eigenfunctions differ from each other. This means that the behavior of the tunneling tail cannot be deduced from the classical dynamics in real phase space.

The discrepancy found here between the map and the corresponding Hamiltonian system is not surprising. As noted in the introduction, even for the same classial Hamiltonian, there is an ambiguity in the quantization arising from the choice of the operator ordering, while the magnitude of the tunneling effects is in general exponentially small with respect to the Planck constant *ℏ*, which cannot be captured by any power series expansion.

A lesson we learned from this example is that it is necessary to investigate the dynamics in the complex plane to understand the tunneling effect properly. Here, we have not performed a fully semiclassical calculation with complex orbits, but it would be reasonable to expect that the difference in the behavior of the complex dynamics gives rise to the difference in the tunneling tail. The branch points, appearing in the potential function of the map, produce the multivaluedness of the complex map whose treatment is one of the topics in the theory of complex dynamical systems [48]. We have shown that the real part of the orbits at the fixed point of the Hamiltonian flow remains the same under the dynamics, while the orbit changes its real part discontinuously before and after crossing the branch point. This must be the origin of the localization of the wave function along the separatrix for the fixed point, but note that this is just a speculation and thus has not been validated yet. Semiclassical analysis remains as future work, so we should perform a single-step semiclassical calculation first since the multi-step semiclassical analysis for the multivalued dynamical system has not been established.

The map $f_{\lambda }$ is slightly complicated compared to the flow $F_{\lambda }$ due to the presence of singularities producing the multivaluedness of the dynamics. This fact reminds us of the so-called Painlevé conjecture: in integrable systems, singularities appearing in the time plane are at most limited to the poles [50]. This, in turn, suggests that the system can be nonintegrable if branch points appear in the complex time plane of the solutions. One can say that the Suris’s integrable map is completely integrable as far as the dynamics in the real plane are concerned, whereas it is one step outside of integrable systems. Our present result shows that this difference is reflected in the tunneling effect.

It is important to note that such subtlety is not limited to the system studied in this paper. If the system is close enough to the integrable limit, the phase space is almost covered by Kolmogorov–Arnold–Moser curves. For the classical-to-quantum correspondence, the size of the Planck cell is an important spatial scale; even if invariant structures such as nonlinear resonances or stochastic layers appear in phase space, it does not matter if their sizes are smaller than the size of the Planck cell under consideration. This is a fundamental working hypothesis when considering the manifestation of classical chaos in the corresponding systems [51]. According to this principle, the nearly integrable system, in which the invariant structures inherent in the nonintegrability are all on the sub-Planck cell scale, should be identified with the completely integrable system.

This must hold true in the real plane, but not necessarily in the complex plane. This is because the phase space profile for the nonintegrable system, no matter how close it is to an integrable system, is dramatically different from that of the completely integrable system. The KAM curves are analytic in the real plane, but they have *natural boundaries* in the complex plane [52,53,54,55]. Furthermore, it is highly probable that homoclinic intersections between stable and unstable manifolds occur beyond the natural boundaries [22]. Chaos in the complex plane developed there is difficult to ‘see’ from the phase space profile in the real plane, but tunneling transport is driven by chaotic orbits in such a deep complex plane [23].

Quantum tunneling is a phenomenon that does not exist in classical mechanics. There are approaches that try to understand quantum tunneling in terms of invariant objects in the real phase space [56,57]. These implicitly assume that tunneling tails of wave functions in nonintegrable systems can be expressed in terms of tunneling tails of wave functions supported by invariant manifolds in the real phase space. In other words, the *broadening* of the wave functions [58] is assumed to be sufficient to capture the signature of nonintegrable tunneling. However, there is a situation in which such an approach does not work [23], and there, one needs to make full use of chaos in the complex plane. Chaos in the complex plane appears beyond natural boundaries and thus is not reachable from the real plane. We believe that this fact is not necessarily limited to the situation where no visible invariant structures appear compared to the Planck cell as reported in Ref. [23], but the mechanism found there works in tunneling processes in nonintegrable systems in general. The phase spaces used so far to study the tunneling in nonintegrable systems are too complex to specify purely quantum phenomena, which could slip through any kind of analysis based on classical dynamics in the real plane. The question of what the unique characteristics of tunneling effects are in nonintegrable systems should be revisited.

## Figures and Tables

**Figure 1 entropy-26-00414-f001:**
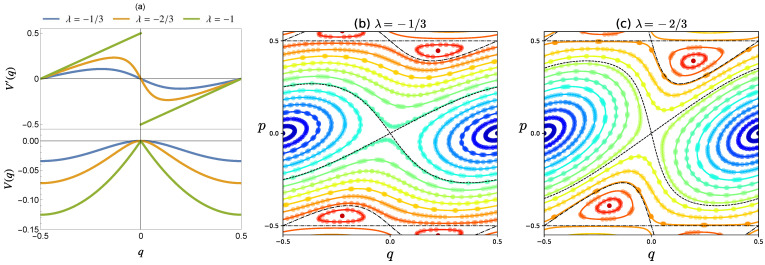
(**a**) Plot of the functions $V_{\lambda }^{\prime }\left(q\right)$ (upper) and $V_{\lambda }\left(q\right)$ (lower) for λ=−1/3, −2/3, and −1. Phase space portrait for (**b**) λ=−1/3 and (**c**) λ=−2/3. In (**b**,**c**), dots and solid curves represent the orbits for $f_{\lambda }$ and the associated contour curve $H_{\lambda }(q,p)$, respectively. Black dashed and black dash-dotted curves represent the separatrix starting from (*q*, *p*) = (0, 0) and (0, ±1/2), respectively.

**Figure 2 entropy-26-00414-f002:**
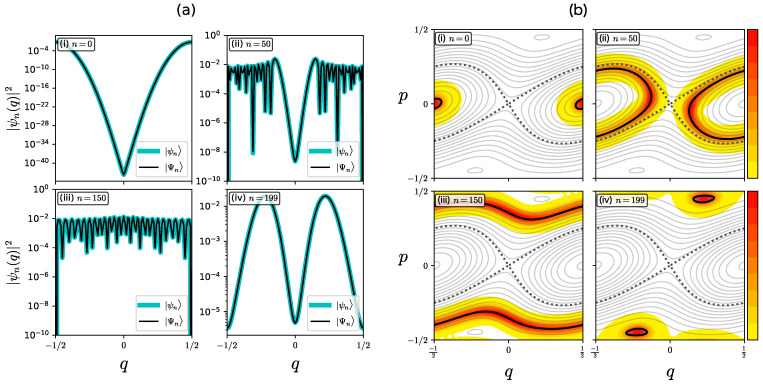
(**a**) Eigenstates $\psi_{n}\left(q\right)$ (cyan) and $\Psi_{n}\left(q\right)$ (black) with ℏ=1/200π for (**i**) n=0, (**ii**) n=50, (**iii**) n=150, and (**iv**) n=199. (**b**) Husimi representation of $\psi_{n}\left(q\right)$ in normal scale for (**i**) n=0, (**ii**) n=50, (**iii**) n=150, and (**iv**) n=199. The intensity of Husim representations is indicated by a yellow-red color scheme shown in the right panel. The black dotted curve shows the contour curve with $H_{\lambda }(q,p)=E_{n}$. The black broken curve shows the separatrix starting from (q,p)=(0,0).

**Figure 3 entropy-26-00414-f003:**
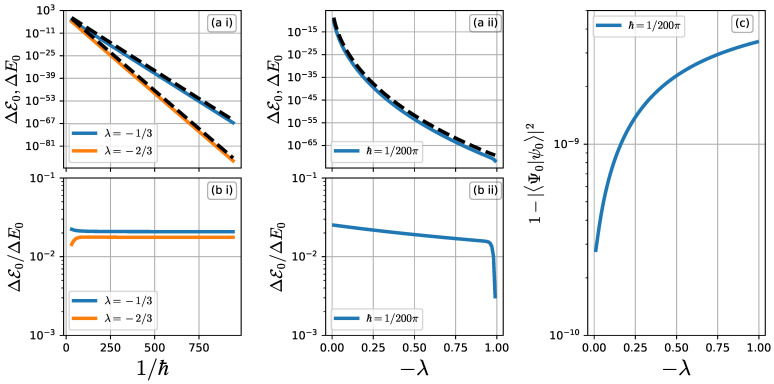
(**a**) Plots of the tunneling splittings $\Delta E_{0}$ (solid) and $\Delta E_{0}$ (dashed) as a function of (**i**) 1/ℏ and (**ii**) −λ. (**b**) Plots of the ratio $\Delta E_{0}/\Delta E_{0}$ as a function (**i**) 1/ℏ and (**ii**) −λ. (**c**) Plot of $1-|\langle \psi_{0}|\Psi_{0}\rangle |$ vs. −λ.

**Figure 4 entropy-26-00414-f004:**
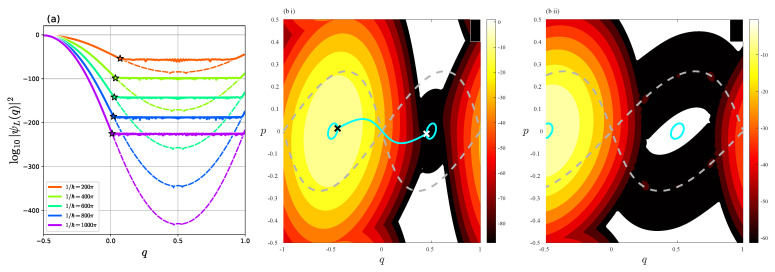
(**a**) Wave functions $\psi_{L}\left(q\right)$ (solid curves) and $\Psi_{L}\left(q\right)$ (dashed curves) for different values of 1/ℏ. The “★” mark is put to indicate the deviation point $q_{-}^{★}$. (**b**) Husimi representation of (**i**) $\Psi_{L}\left(q\right)$ and (**ii**) $\psi_{L}\left(q\right)$ for 1/ℏ=200π in the $\log_{10}$ scale. The dashed gray curve represents the separatrix. Closed cyan curves show the contour curve associated with the ground state energy level $E_{0}=-1.63905702$. Black and white “×” symbols indicate the position of the turning points for q(t); (q,p)=(±0.450847,±0.012215), respectively. (**i**) The cyan curve connected to the two turning points shows the instanton curve projected onto the (Req,Rep)-plane.

**Figure 5 entropy-26-00414-f005:**
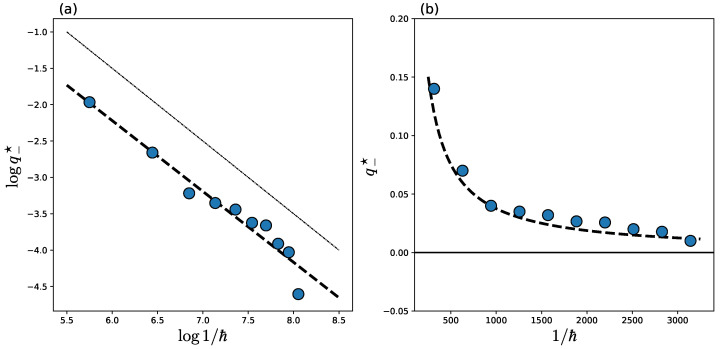
The dots represent the deviation point $q_{-}^{★}$ vs. 1/ℏ for λ=−1/3 on (**a**) the double logarithmic scale and (**b**) the normal scale. The dashed line is obtained by applying linear regression to the plot of (**a**) whose slope is obtained as −0.97418081. The dotted-dashed line with slope −1 in (**a**) is shown just for reference. The black horizontal line in (**b**) represents $q_{-}^{★}=0$ as a guide.

**Figure 6 entropy-26-00414-f006:**
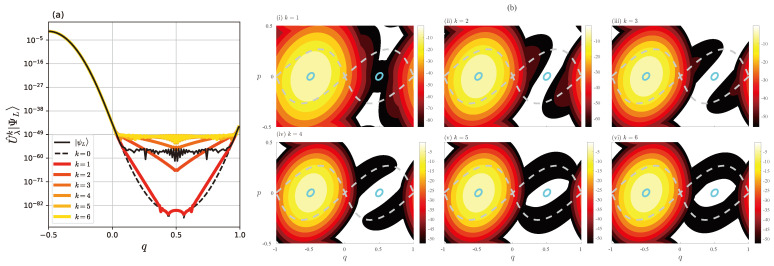
Plot of ${\hat{U}}_{\lambda }^{k}|\Psi_{L}\rangle$ (**a**) in the *q*-representation and (**b**) in the Husimi representation. The dashed gray curve in (**b**) represents the separatrix starting from (q,p)=(0,0).

**Figure 7 entropy-26-00414-f007:**
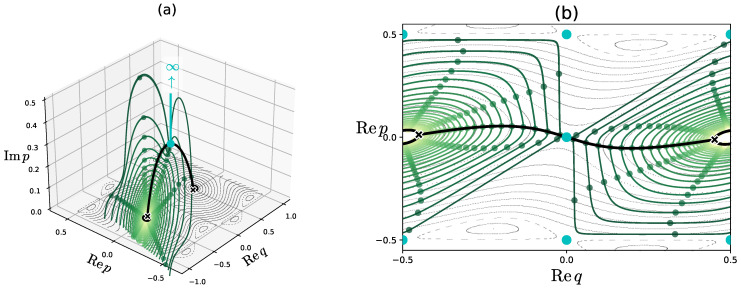
The solution curves obtained under the continuous Hamiltonian flow $F_{\lambda }$. They are confined in the equi-energy surface given by the condition $H_{\lambda }=E_{0}=-1.63905702$. The curves are projected onto (**a**) the (Req,Rep,Imp)-space and (**b**) the (Req,Rep)-plane. In (**a**,**b**) the black closed curve and the black semicircular curve represent the equi-energy curves in the real plane and the instanton curve, respectively. The symbol “×” stands for the turning point of Req(t). (**a**) The cyan line tending to infinity shows a solution curve of $H_{\lambda }$ starting from the point (Req,Rep)=(0,0), marked by the cyan point on the instanton. (**b**) The cyan dots show the orbits of $f_{\lambda }$ starting from the point (Req,Rep)=(0,0) on the instanton.

**Figure 8 entropy-26-00414-f008:**
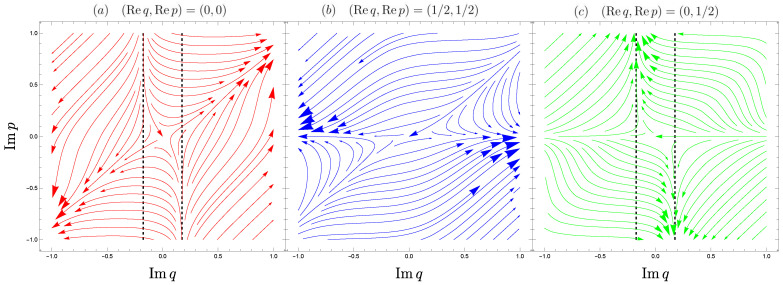
Flow of Equation (13) in the (Imq,Imp)-plane for λ=−1/3. Vertical dashed lines represent Imq=q⋄ for λ=−1/3.

**Figure 9 entropy-26-00414-f009:**
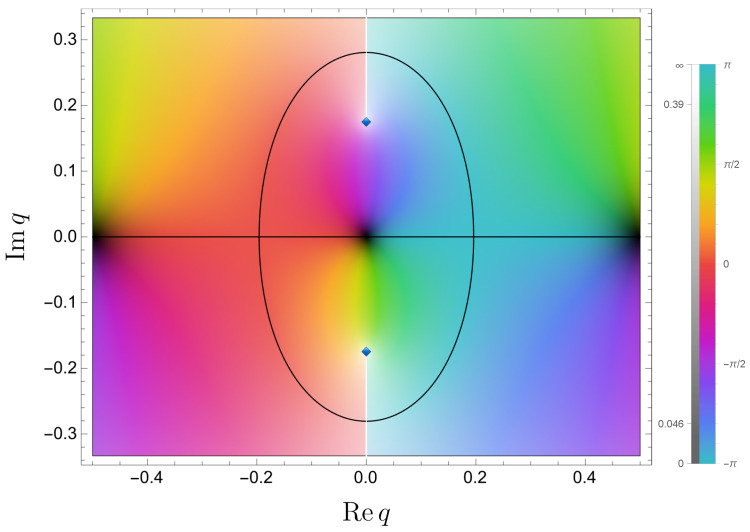
The function $V_{\lambda }^{\prime }\left(q\right)$ in the (Req,Imq)-plane with λ=−1/3. The argument of $V_{\lambda }^{\prime }\left(q\right)$ and the absolute value $|V_{\lambda }^{\prime }\left(q\right)|$ are distinguished by color and brightness, respectively. The blue diamond “⋄” symbol represents the branch point of $V^{\prime }\left(q\right)$. The white line starting from the branch point represents the branch cut. The black curve shows a contour curve ImVλ′(q)=0.

**Figure 10 entropy-26-00414-f010:**
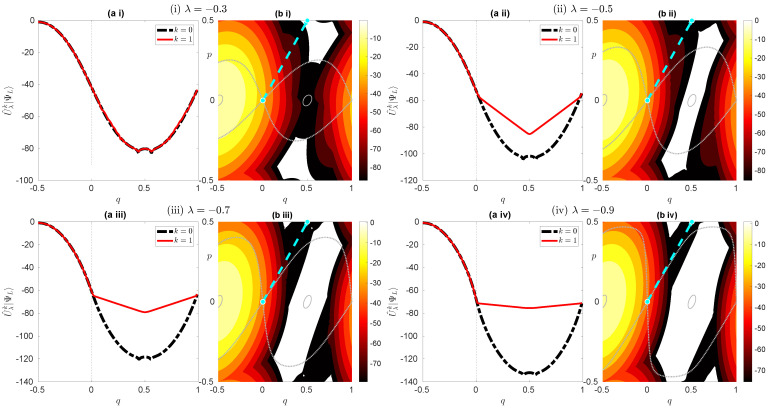
(**a**) Plot of the wave functions ${\hat{U}}_{\lambda }|\Psi_{L}\rangle$ (red curve) and $|\Psi_{L}\rangle$ (black dashed curve) for (**i**) λ=−0.3, (**ii**) λ=−0.5, (**iii**) λ=−0.7, and (**iv**) λ=−0.9. (**b**) Husimi representation of ${\hat{U}}_{\lambda }|\Psi_{L}\rangle$. The cyan dotted line represents a line (q,p)=(0,0) to (q,p)=(1/2,1/2) for a guide. The gray solid curve and the gray dashed curve represent a contour curve of $H_{\lambda }(q,p)=E_{0}$ and the separatrix starting from (q,p)=(0,0), respectively.

## Data Availability

The data presented in this study are available on request from the corresponding author.

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
