# Peer review of "Quantum Tunneling and Complex Dynamics in the Suris’s Integrable Map"

_entropy, 2024, doi:10.3390/e26050414_

Round 1

Reviewer 1 Report

Comments and Suggestions for Authors

Tunneling is studied for a 2-dimensional integrable map introduced by Suris. The results found for quantum tunneling for an integrable map are compared with tunneling found for the associated Hamiltonian. The authors find interesting properties of the tunneling tails in the eigenfunctions. In particular the tunneling tail cannot be captured by classical dynamics in real phase space, as is often done, pointing to the importance of studying the dynamics in the complex plane, which is analyzed in detail for this system. The authors explain that this is to be expected in many systems, not just the one that is the focus of the study presented in the manuscript. The manuscript will interest many readers of Entropy and publication is recommended after the authors address the following:

On page 5 the authors discuss the tunneling splittings plotted in figure 3. Though if is apparent that there is a difference between the pair of tunneling splittings in the figure, the splittings themselves are extremely small, and from the curves we cannot tell what the difference is. The authors refer to “a few orders of magnitude difference”. While certainly plausible from inspection of the figure, perhaps the authors might give some examples.

Author Response

We are grateful for the careful reading of our manuscript and for giving constructive comments.The following are the replies to the referee: To clarify the ``a few order of magnitude difference", we have added new figures [Figures 3(b i) and 3(b ii)], and revised the text in line 176 on page 6. We hope that this revision will improve the readability of our manuscript. 

Reviewer 2 Report

Comments and Suggestions for Authors

My understanding of the paper is the following. The authors study a classical map and its corresponding Hamiltonian system, along with their quantized counterparts. The authors numerically show there is a difference in the tail of the wave function of the quantum map as compared to the tail of the wave function of the Hamiltonian system. They attribute the difference to quantum tunneling in the quantum map, which much more pronounced  than in the Hamiltonian system. The authors propose that the enhancement of quantum tunneling in the quantum map is connected to a branch-point singularity of derivative of the potential energy in the complex position plane, which appears in the classical map, but does not appear in the classical Hamiltonian system.  The author suggest that singularities in the complex plane like the one they studied could play a role in the dynamics of other systems, including chaotic systems. 

If my understanding is correct, the authors are proposing a conjecture, namely that a complex singularity of the potential in the classical map is connected to tunneling in the corresponding quantum map. As far as I could see the authors didn't present any strong evidence for this claim. This is OK as long as the authors clearly say that they are proposing a conjecture. My main suggestion is to say this clearly both in the introduction and in the discussion section. 

In addition, I have a few other  questions and suggestions:

1) if I understood correctly, in the Husimi plots  the color scale is the logarithm of the  absolute value squared of the wave function in the Husimi representation. If this is correct, please add this statement to the figure captions. 

2) In Figures 3a and 3b add "Delta E_0" to the vertical axis. 

3) In lines 311-331 the authors discuss some difficulties that prevent them from reaching a firm conclusion on the role of the branch point on the tunneling along the separatrix seen in the quantum map. The authors also state possible ways to overcome some of these difficulties. I suggest the authors state clearly that these difficulties should be addressed in future studies. 

4) The tails of the wave functions studied by the authors are incredibly small, so one should be careful to distinguish the tunneling effect from numerical errors. In basic quantum tunneling, the tunneling effect can be significantly enhanced by reducing  the height or width of the tunneling barrier. Can the authors do this? Would the difference in the tails of the wave functions remain when the tunneling effect is strong?

5) What is the limit of the tunneling splitting (figure 3) when lambda goes to zero? I am a little confused as to why there seems to be a splitting even when lambda is zero. 

I will recommend publication of the paper after the authors have addressed these points.

Comments on the Quality of English Language

1) In line 32 it says "Since only energetic barrier tunneling but also dynamical tunneling..." I think it should say "Since not only energetic barrier tunneling but also dynamical tunneling..."

2) The authors refer to their paper as "report". I think is better to just call it "paper".

3) In line 257 it says "greens dots". It should say "green dots". 

Author Response

We are grateful for the careful reading of our manuscript and for giving constructive comments. Please find attached our response to the referee.

Reviewer 3 Report

Comments and Suggestions for Authors

This paper provides a thorough study of quantum tunneling within an integrable 2D map, using analytical continuation of the classical dynamics to the complex domain. The approach and the results are compared to the corresponding continuous-time one-degree-of-freedom Hamiltonian, which displays very similar classical phase-space structures and very similar Husimi densities of eigenstates as the quantum map. Tiny differences in the tunneling rate are traced back to subtle but rather fundamental differences in the complex phase space when comparing the map to the continuous-time system.

I can recommend this paper for publication in Entropy, even though the study is sort of unfinished as the authors point out themselves in the manuscript. Indeed, the fact that an integrable symplectic 2D map is not exactly the same thing as a conservative one-degree-of-freedom Hamiltonian system represents an information that is of importance for the scientific community dealing with the semiclassical theory of tunneling. I have a few observations and comments that the authors might want to take into account:

1) All viewgraphs show detailed information that support the conclusions in the text. For a few of them, I think it makes sense to show some complementary representations. In particular:
a) In Fig. 3 (a) and (b) I would be curious to see the ratio of splittings, for the map and the Hamiltonian, as a function of 1/\hbar and of -\lambda. This would give more quantitative insight into the discrepancy between these two systems.
b) In Fig. 5 a complementary log-log representation could be useful and give more insight.

2) in Fig. 4(a) the non-smooth behaviour of the left function associated with the map is rather intriguing. Can this be related to the map character of the corresponding quantum system? A brief comment here would be useful.

3) In Lines 220 and 221, it is written that \ket{\Psi_L} is an eigenstate of the Hamiltonian \hat{H}_\lambda. As far as I understand, this is not exactly true since \Psi_L is created by a superposition of two eigenstates with slightly different energies, according to Eq. (10). This statement should thus be corrected.

4) While reading the caption of Fig. 4, I did not (yet) immediately understand what (x,u) stand for. I would thus recommend to replace here x by Re(q) and u by Re(p), to make this caption self-understandable.

5) Minor spelling issue in Line 361: "Poinlevè" should read "Painlevé".

Comments on the Quality of English Language

The quality of English is very good in this manuscript. I detected very few issues (e.g. "The greens dots ...", l. 257) to be corrected.

Author Response

(The authors gave the same response as above.)

Round 2

Reviewer 2 Report

Comments and Suggestions for Authors

I thank the authors for answering my questions. I have a better understanding of their results now. I recommend the paper be published after minor English editing. 

Comments on the Quality of English Language

The paper requires minor editing, but the Authors's English is good enough for publication.

Reviewer 3 Report

Comments and Suggestions for Authors

The authors responded to my comments in a satisfactory manner. I thus recommend to publish this paper in its present form.

Comments on the Quality of English Language

The quality of the English language is very good. I detected a few typos ("The greens dots ..." -> "The green dots ...", "Poinlevé" -> "Painlevé", "with slop" -> "with slope", ...) that ought to be corrected before publication.